# NEWSQA: A MACHINE COMPREHENSION DATASET

**Adam Trischler**[*]       **Tong Wang**[*]       **Xingdi Yuan**[*]       **Justin Harris**

**Alessandro Sordoni**       **Philip Bachman**       **Kaheer Suleman**

{adam.trischler, tong.wang, eric.yuan, justin.harris,
 alessandro.sordoni, phil.bachman, k.suleman}@maluuba.com
Maluuba Research
Montréal, Québec, Canada

## ABSTRACT

We present *NewsQA*, a challenging machine comprehension dataset of over 100,000 question-answer pairs. Crowdworkers supply questions and answers based on a set of over 10,000 news articles from CNN, with answers consisting in spans of text from the corresponding articles. We collect this dataset through a four-stage process designed to solicit exploratory questions that require reasoning. A thorough analysis confirms that *NewsQA* demands abilities beyond simple word matching and recognizing entailment. We measure human performance on the dataset and compare it to several strong neural models. The performance gap between humans and machines (0.198 in F1) indicates that significant progress can be made on *NewsQA* through future research. The dataset is freely available at datasets.maluuba.com/NewsQA.

## 1 INTRODUCTION

Almost all human knowledge is recorded in the language of text. As such, comprehension of written language by machines, at a near-human level, would enable a broad class of artificial intelligence applications. In human students we evaluate reading comprehension by posing questions based on a text passage and then assessing a student's answers. Such comprehension tests are appealing because they are objectively gradable and may measure a range of important abilities, from basic understanding to causal reasoning to inference (Richardson et al., 2013). To teach literacy to machines, the research community has taken a similar approach with machine comprehension (MC).

Recent years have seen the release of a host of MC datasets. Generally, these consist of (document, question, answer) triples to be used in a supervised learning framework. Existing datasets vary in size, difficulty, and collection methodology; however, as pointed out by Rajpurkar et al. (2016), most suffer from one of two shortcomings: those that are designed explicitly to test comprehension (Richardson et al., 2013) are too small for training data-intensive deep learning models, while those that are sufficiently large for deep learning (Hermann et al., 2015; Hill et al., 2016; Bajgar et al., 2016) are generated synthetically, yielding questions that are not posed in natural language and that may not test comprehension directly (Chen et al., 2016). More recently, Rajpurkar et al. (2016) sought to overcome these deficiencies with their crowdsourced dataset, *SQuAD*.

Here we present a challenging new largescale dataset for machine comprehension: *NewsQA*. *NewsQA* contains 119,633 natural language questions posed by crowdworkers on 12,744 news articles from CNN. Answers to these questions consist in spans of text within the corresponding article highlighted by a distinct set of crowdworkers. To build *NewsQA* we utilized a four-stage collection process designed to encourage exploratory, curiosity-based questions that reflect human information seeking. CNN articles were chosen as the source material because they have been used in the past (Hermann et al., 2015) and, in our view, machine comprehension systems are particularly suited to high-volume, rapidly changing information sources like news.

---

[*]These three authors contributed equally.

As Trischler et al. (2016a), Chen et al. (2016), and others have argued, it is important for datasets to be sufficiently challenging to teach models the abilities we wish them to learn. Thus, in line with Richardson et al. (2013), our goal with *NewsQA* was to construct a corpus of questions that necessitates reasoning mechanisms, such as synthesis of information across different parts of an article. We designed our collection methodology explicitly to capture such questions.

The challenging characteristics of *NewsQA* that distinguish it from most previous comprehension tasks are as follows:

1. Answers are spans of arbitrary length within an article, rather than single words or entities.

2. Some questions have no answer in the corresponding article (the *null* span).

3. There are no candidate answers from which to choose.

4. Our collection process encourages lexical and syntactic divergence between questions and answers.

5. A significant proportion of questions requires reasoning beyond simple word- and context-matching (as shown in our analysis).

In this paper we describe the collection methodology for *NewsQA*, provide a variety of statistics to characterize it and contrast it with previous datasets, and assess its difficulty. In particular, we measure human performance and compare it to that of two strong neural-network baselines. Unsurprisingly, humans significantly outperform the models we designed and assessed, achieving an F1 score of 0.694 versus 0.496 for the best-performing machine. We hope that this corpus will spur further advances on the challenging task of machine comprehension.

## 2 RELATED DATASETS

*NewsQA* follows in the tradition of several recent comprehension datasets. These vary in size, difficulty, and collection methodology, and each has its own distinguishing characteristics. We agree with Bajgar et al. (2016) who have said "models could certainly benefit from as diverse a collection of datasets as possible." We discuss this collection below.

### 2.1 MCTEST

*MCTest* (Richardson et al., 2013) is a crowdsourced collection of 660 elementary-level children's stories with associated questions and answers. The stories are fictional, to ensure that the answer must be found in the text itself, and carefully limited to what a young child can understand. Each question comes with a set of 4 candidate answers that range from single words to full explanatory sentences. The questions are designed to require rudimentary reasoning and synthesis of information across sentences, making the dataset quite challenging. This is compounded by the dataset's size, which limits the training of expressive statistical models. Nevertheless, recent comprehension models have performed well on *MCTest* (Sachan et al., 2015; Wang et al., 2015), including a highly structured neural model (Trischler et al., 2016a). These models all rely on access to the small set of candidate answers, a crutch that *NewsQA* does not provide.

### 2.2 CNN/DAILY MAIL

The *CNN/Daily Mail* corpus (Hermann et al., 2015) consists of news articles scraped from those outlets with corresponding cloze-style questions. Cloze questions are constructed synthetically by deleting a single entity from abstractive summary points that accompany each article (written presumably by human authors). As such, determining the correct answer relies mostly on recognizing textual entailment between the article and the question. The named entities within an article are identified and anonymized in a preprocessing step and constitute the set of candidate answers; contrast this with *NewsQA* in which answers often include longer phrases and no candidates are given.

Because the cloze process is automatic, it is straightforward to collect a significant amount of data to support deep-learning approaches: *CNN/Daily Mail* contains about 1.4 million question-answer pairs. However, Chen et al. (2016) demonstrated that the task requires only limited reasoning and, in

fact, performance of the strongest models (Kadlec et al., 2016; Trischler et al., 2016b; Sordoni et al., 2016) nearly matches that of humans.

## 2.3 CHILDREN'S BOOK TEST

The *Children's Book Test* (*CBT*) (Hill et al., 2016) was collected using a process similar to that of *CNN/Daily Mail*. Text passages are 20-sentence excerpts from children's books available through Project Gutenberg; questions are generated by deleting a single word in the next (*i.e.*, 21st) sentence. Consequently, *CBT* evaluates word prediction based on context. It is a comprehension task insofar as comprehension is likely necessary for this prediction, but comprehension may be insufficient and other mechanisms may be more important.

## 2.4 BOOKTEST

Bajgar et al. (2016) convincingly argue that, because existing datasets are not large enough, we have yet to reach the full capacity of existing comprehension models. As a remedy they present *BookTest*. This is an extension to the named-entity and common-noun strata of *CBT* that increases their size by over 60 times. Bajgar et al. (2016) demonstrate that training on the augmented dataset yields a model (Kadlec et al., 2016) that matches human performance on *CBT*. This is impressive and suggests that much is to be gained from more data, but we repeat our concerns about the relevance of story prediction as a comprehension task. We also wish to encourage more efficient learning from less data.

## 2.5 SQUAD

The comprehension dataset most closely related to *NewsQA* is *SQuAD* (Rajpurkar et al., 2016). It consists of natural language questions posed by crowdworkers on paragraphs from high-PageRank Wikipedia articles. As in *NewsQA*, each answer consists of a span of text from the related paragraph and no candidates are provided. Despite the effort of manual labelling, *SQuAD*'s size is significant and amenable to deep learning approaches: 107,785 question-answer pairs based on 536 articles.

*SQuAD* is a challenging comprehension task in which humans far outperform machines. The authors measured human accuracy at 0.905 in F1 (we measured human F1 at 0.807 using a different methodology), whereas at the time of the writing, the strongest published model to date achieves only 0.700 in F1 (Wang & Jiang, 2016b).

## 3 COLLECTION METHODOLOGY

We collected *NewsQA* through a four-stage process: article curation, question sourcing, answer sourcing, and validation. We also applied a post-processing step with answer agreement consolidation and span merging to enhance the usability of the dataset.

### 3.1 ARTICLE CURATION

We retrieve articles from CNN using the script created by Hermann et al. (2015) for *CNN/Daily Mail*. From the returned set of 90,266 articles, we select 12,744 uniformly at random. These cover a wide range of topics that includes politics, economics, and current events. Articles are partitioned at random into a training set (90%), a development set (5%), and a test set (5%).

### 3.2 QUESTION SOURCING

It was important to us to collect challenging questions that could not be answered using straightforward word- or context-matching. Like Richardson et al. (2013) we want to encourage reasoning in comprehension models. We are also interested in questions that, in some sense, model human curiosity and reflect actual human use-cases of information seeking. Along a similar line, we consider it an important (though as yet overlooked) capacity of a comprehension model to recognize when given information is inadequate, so we are also interested in questions that may not have sufficient evidence in the text. Our question sourcing stage was designed to solicit questions of this nature, and deliberately separated from the answer sourcing stage for the same reason.

*Questioners* (a distinct set of crowdworkers) see *only* a news article's headline and its summary points (also available from CNN); they do not see the full article itself. They are asked to formulate a question from this incomplete information. This encourages curiosity about the contents of the full article and prevents questions that are simple reformulations of sentences in the text. It also increases the likelihood of questions whose answers do not exist in the text. We reject questions that have significant word overlap with the summary points to ensure that crowdworkers do not treat the summaries as mini-articles, and further discouraged this in the instructions. During collection each Questioner is solicited for up to three questions about an article. They are provided with positive and negative examples to prompt and guide them (detailed instructions are shown in Figure 3).

### 3.3 ANSWER SOURCING

A second set of crowdworkers (*Answerers*) provide answers. Although this separation of question and answer increases the overall cognitive load, we hypothesized that unburdening Questioners in this way would encourage more complex questions. Answerers receive a full article along with a crowdsourced question and are tasked with determining the answer. They may also reject the question as nonsensical, or select the *null* answer if the article contains insufficient information. Answers are submitted by clicking on and highlighting words in the article while instructions encourage the set of answer words to consist in a single continuous span (again, we give an example prompt in the Appendix). For each question we solicit answers from multiple crowdworkers (avg. 2.73) with the aim of achieving agreement between at least two Answerers.

### 3.4 VALIDATION

Crowdsourcing is a powerful tool but it is not without peril (collection glitches; uninterested or malicious workers). To obtain a dataset of the highest possible quality we use a validation process that mitigates some of these issues. In validation, a third set of crowdworkers sees the full article, a question, and the set of unique answers to that question. We task these workers with choosing the best answer from the candidate set or rejecting all answers. Each article-question pair is validated by an average of 2.48 crowdworkers. Validation was used on those questions without answer-agreement after the previous stage, amounting to 43.2% of all questions.

### 3.5 ANSWER MARKING AND CLEANUP

After validation, 86.0% of all questions in *NewsQA* have answers agreed upon by at least two separate crowdworkers—either at the initial answer sourcing stage or in the top-answer selection. This improves the dataset's quality. We choose to include the questions without agreed answers in the corpus also, but they are specially marked. Such questions could be treated as having the *null* answer and used to train models that are aware of poorly posed questions.

As a final cleanup step we combine answer spans that are less than 3 words apart (punctuation is discounted). We find that 5.68% of answers consist in multiple spans, while 71.3% of multi-spans are within the 3-word threshold. Looking more closely at the data reveals that the multi-span answers often represent lists. These may present an interesting challenge for comprehension models moving forward.

## 4 DATA ANALYSIS

We provide a thorough analysis of *NewsQA* to demonstrate its challenge and its usefulness as a machine comprehension benchmark. The analysis focuses on the types of answers that appear in the dataset and the various forms of reasoning required to solve it.[1]

### 4.1 ANSWER TYPES

Following Rajpurkar et al. (2016), we categorize answers based on their linguistic type (see Table 1). This categorization relies on Stanford CoreNLP to generate constituency parses, POS tags, and NER

---

[1]Additional statistics are available at `http://datasets.maluuba.com/NewsQA/stats`.

Table 1: The variety of answer types appearing in *NewsQA*, with proportion statistics and examples.

| Answer type | Example | Proportion (%) |
|---|---|---|
| Date/Time | March 12, 2008 | 2.9 |
| Numeric | 24.3 million | 9.8 |
| Person | Ludwig van Beethoven | 14.8 |
| Location | Torrance, California | 7.8 |
| Other Entity | Pew Hispanic Center | 5.8 |
| Common Noun Phrase | federal prosecutors | 22.2 |
| Adjective Phrase | 5-hour | 1.9 |
| Verb Phrase | suffered minor damage | 1.4 |
| Clause Phrase | trampling on human rights | 18.3 |
| Prepositional Phrase | in the attack | 3.8 |
| Other | nearly half | 11.2 |

tags for answer spans (see Rajpurkar et al. (2016) for more details). From the table we see that the majority of answers (22.2%) are common noun phrases. Thereafter, answers are fairly evenly spread among the clause phrase (18.3%), person (14.8%), numeric (9.8%), and other (11.2%) types. Clearly, answers in *NewsQA* are linguistically diverse.

The proportions in Table 1 only account for cases when an answer span exists. The complement of this set comprises questions with an agreed *null* answer (9.5% of the full corpus) and answers without agreement after validation (4.5% of the full corpus).

## 4.2 REASONING TYPES

The forms of reasoning required to solve *NewsQA* directly influence the abilities that models will learn from the dataset. We stratified reasoning types using a variation on the taxonomy presented by Chen et al. (2016) in their analysis of the *CNN/Daily Mail* dataset. Types are as follows, in ascending order of difficulty:

1. **Word Matching:** Important words in the question exactly match words in the immediate context of an answer span such that a keyword search algorithm could perform well on this subset.

2. **Paraphrasing:** A single sentence in the article entails or paraphrases the question. Paraphrase recognition may require synonymy and word knowledge.

3. **Inference:** The answer must be inferred from incomplete information in the article or by recognizing conceptual overlap. This typically draws on world knowledge.

4. **Synthesis:** The answer can only be inferred by synthesizing information distributed across multiple sentences.

5. **Ambiguous/Insufficient:** The question has no answer or no unique answer in the article.

For both *NewsQA* and *SQuAD*, we manually labelled 1,000 examples (drawn randomly from the respective development sets) according to these types and compiled the results in Table 2. Some examples fall into more than one category, in which case we defaulted to the more challenging type. We can see from the table that word matching, the easiest type, makes up the largest subset in both datasets (32.7% for *NewsQA* and 39.8% for *SQuAD*). Paraphrasing constitutes a much larger proportion in *SQuAD* than in *NewsQA* (34.3% vs 27.0%), possibly a result from the explicit encouragement of lexical variety in *SQuAD* question sourcing. However, *NewsQA* significantly outnumbers *SQuAD* on the distribution of the more difficult forms of reasoning: synthesis and inference make up 33.9% of the data in contrast to 20.5% in *SQuAD*.

## 5 BASELINE MODELS

We test the performance of three comprehension systems on *NewsQA*: human data analysts and two neural models. The first neural model is the match-LSTM (mLSTM) system of Wang & Jiang

Table 2: Reasoning mechanisms needed to answer questions. For each we show an example question with the sentence that contains the answer span, with words relevant to the reasoning type in **bold**, and the corresponding proportion in the human-evaluated subset of both *NewsQA* and *SQuAD* (1,000 samples each).

| Reasoning | Example | Proportion (%) | |
|---|---|---|---|
| | | *NewsQA* | *SQuAD* |
| Word Matching | Q: **When were** the **findings published**? <br> S: Both sets of research **findings were published Thursday**... | 32.7 | 39.8 |
| Paraphrasing | Q: **Who** is the **struggle between** in Rwanda? <br> S: The **struggle pits ethnic Tutsis**, supported by Rwanda, **against ethnic Hutu**, backed by Congo. | 27.0 | 34.3 |
| Inference | Q: **Who** drew **inspiration** from **presidents**? <br> S: **Rudy Ruiz** says the lives of US **presidents** can make them **positive role models** for students. | 13.2 | 8.6 |
| Synthesis | Q: **Where** is Brittanee **Drexel** from? <br> S: The mother of a 17-year-old **Rochester**, **New York** high school student ... says she did not give her daughter permission to go on the trip. **Brittanee** Marie **Drexel**'s mom says... | 20.7 | 11.9 |
| Ambiguous/Insufficient | Q: **Whose mother** is **moving** to the White House? <br> S: ... **Barack Obama's mother-in-law**, Marian Robinson, will **join** the Obamas at the **family's private quarters** at 1600 Pennsylvania Avenue. [Michelle is never mentioned] | 6.4 | 5.4 |

(2016b). The second is a model of our own design that is computationally cheaper. We describe these models below but omit the personal details of our analysts. Implementation details of the models are described in Appendix A.

## 5.1 MATCH-LSTM

There are three stages involved in the mLSTM model. First, LSTM networks encode the document and question (represented by GloVe word embeddings (Pennington et al., 2014)) as sequences of hidden states. Second, an mLSTM network (Wang & Jiang, 2016a) compares the document encodings with the question encodings. This network processes the document sequentially and at each token uses an attention mechanism to obtain a weighted vector representation of the question; the weighted combination is concatenated with the encoding of the current token and fed into a standard LSTM. Finally, a Pointer Network uses the hidden states of the mLSTM to select the boundaries of the answer span. We refer the reader to Wang & Jiang (2016a;b) for full details. At the time of writing, mLSTM is state-of-the-art on *SQuAD* (see Table 3) so it is natural to test it further on *NewsQA*.

## 5.2 THE BILINEAR ANNOTATION RE-ENCODING BOUNDARY (BARB) MODEL

The match-LSTM is computationally intensive since it computes an attention over the entire question at each document token in the recurrence. To facilitate faster experimentation with *NewsQA* we developed a lighter-weight model (BARB) that achieves similar results on *SQuAD*[2]. Our model consists in four stages:

**Encoding** All words in the document and question are mapped to real-valued vectors using the GloVe embedding matrix $\mathbf{W} \in \mathbb{R}^{|V| \times d}$. This yields $\mathbf{d}_1, \ldots, \mathbf{d}_n \in \mathbb{R}^d$ and $\mathbf{q}_1, \ldots, \mathbf{q}_m \in \mathbb{R}^d$. A bidirectional GRU network (Bahdanau et al., 2015) takes in $\mathbf{d}_i$ and encodes contextual states $\mathbf{h}_i \in \mathbb{R}^{D_1}$ for the document. The same encoder is applied to $\mathbf{q}_j$ to derive contextual states $\mathbf{k}_j \in \mathbb{R}^{D_1}$ for the question.[3]

**Bilinear Annotation** Next we compare the document and question encodings using a set of $C$ bilinear transformations,

$$\mathbf{g}_{ij} = \mathbf{h}_i^T \mathbf{T}^{[1:C]} \mathbf{k}_j, \quad \mathbf{T}^c \in \mathbb{R}^{D_1 \times D_1}, \ \mathbf{g}_{ij} \in \mathbb{R}^C,$$

which we use to produce an $(n \times m \times C)$-dimensional tensor of annotation scores, $\mathbf{G} = [\mathbf{g}_{ij}]$. We take the maximum over the question-token (second) dimension and call the columns of the resulting

---

[2]With the configurations for the results reported in Section 6.2, one epoch of training on *NewsQA* takes about 3.9k seconds for *BARB* and 8.1k seconds for *mLSTM*.

[3]A bidirectional GRU concatenates the hidden states of two GRU networks running in opposite directions. Each of these has hidden size $\frac{1}{2} D_1$.

matrix $\mathbf{g}_i \in \mathbb{R}^C$. We use this matrix as an annotation over the document word dimension. Contrasting the multiplicative application of attention vectors, this annotation matrix is to be concatenated to the encoder RNN input in the re-encoding stage.

**Re-encoding**    For each document word, the input of the re-encoding RNN (another biGRU network) consists of three components: the document encodings $\mathbf{h_i}$, the annotation vectors $\mathbf{g_i}$, and a binary feature $q_i$ indicating whether the document word appears in the question. The resulting vectors $\mathbf{f}_i = [\mathbf{h}_i; \mathbf{g}_i; q_i]$ are fed into the re-encoding RNN to produce $D_2$-dimensional encodings $\mathbf{e}_i$ as input in the boundary-pointing stage.

**Boundary pointing**    Finally, we search for the boundaries of the answer span using a convolutional network (in a process similar to edge detection). Encodings $\mathbf{e}_i$ are arranged in matrix $\mathbf{E} \in \mathbb{R}^{D_2 \times n}$. $\mathbf{E}$ is convolved with a bank of $n_f$ filters, $\mathbf{F}_k^\ell \in \mathbb{R}^{D_2 \times w}$, where $w$ is the filter width, $k$ indexes the different filters, and $\ell$ indexes the layer of the convolutional network. Each layer has the same number of filters of the same dimensions. We add a bias term and apply a nonlinearity (ReLU) following each convolution, with the result an $(n_f \times n)$-dimensional matrix $\mathbf{B}_\ell$.

We use two convolutional layers in the boundary-pointing stage. Given $\mathbf{B}_1$ and $\mathbf{B}_2$, the answer span's start- and end-location probabilities are computed using $p(s) \propto \exp\left(\mathbf{v}_s^T \mathbf{B}_1 + b_s\right)$ and $p(e) \propto \exp\left(\mathbf{v}_e^T \mathbf{B}_2 + b_e\right)$, respectively. We also concatenate $p(s)$ to the input of the second convolutional layer (along the $n_f$-dimension) so as to condition the end-boundary pointing on the start-boundary. Vectors $\mathbf{v}_s, \mathbf{v}_e \in \mathbb{R}^{n_f}$ and scalars $b_s, b_e \in \mathbb{R}$ are trainable parameters.

We also provide an intermediate level of "guidance" to the annotation mechanism by first reducing the feature dimension $C$ in $\mathbf{G}$ with mean-pooling, then maximizing the softmax probabilities in the resulting ($n$-dimensional) vector corresponding to the answer word positions in each document. This auxiliary task is observed empirically to improve performance.

# 6    EXPERIMENTS[4]

## 6.1    HUMAN EVALUATION

We tested four English speakers (three native and one near-native) on a total of 1,000 questions from the *NewsQA* development set. As given in Table 3, they averaged 0.694 in F1, which likely represents a ceiling for machine performance. Our students' exact match (EM) scores are relatively low at 0.465. This is because in many cases there are multiple ways to select semantically equivalent answers, *e.g.*, "1996" versus "in 1996". We also compared human performance on the answers that had agreement with and without validation, finding a difference of only 1.4 percentage points F1. This suggests our validation stage yields good-quality answers.

The original *SQuAD* evaluation of human performance compares separate answers given by crowd-workers; for a closer comparison with *NewsQA*, we replicated our human test on the same number of validation data (1,000). We measured their answers against the second group of crowdsourced responses in *SQuAD*'s development set, as in Rajpurkar et al. (2016). Our students scored 0.807 in F1.

## 6.2    MODEL PERFORMANCE

Performance of the baseline models and humans is measured by EM and F1 with the official evaluation script from *SQuAD* and listed in Table 3. Unless otherwise stated, hyperparameters are determined by `hyperopt` (Appendix A). The gap between human and machine performance on *NewsQA* is a striking 0.198 points F1 — much larger than the gap on *SQuAD* (0.098) under the same human evaluation scheme. The gaps suggest a large margin for improvement with automated methods.

Figure 1 stratifies model (BARB) performance according to answer type (left) and reasoning type (right) as defined in Sections 4.1 and 4.2, respectively. The answer-type stratification suggests that

---

[4]All experiments in this section use the subset of *NewsQA* dataset with answer agreements (92,549 samples for training, 5,166 for validation, and 5,126 for testing). We leave the challenge of identifying the unanswerable questions for future work.

Table 3: Performance of several methods and humans on the *SQuAD* and *NewsQA* datasets. Superscript 1 indicates the results are taken from Rajpurkar et al. (2016), and 2 from Wang & Jiang (2016b).

| *SQuAD* | Exact Match | | F1 | |
| --- | --- | --- | --- | --- |
| Model | Dev | Test | Dev | Test |
| Random[1] | 0.11 | 0.13 | 0.41 | 0.43 |
| mLSTM[2] | 0.591 | 0.595 | 0.700 | 0.703 |
| BARB | 0.591 | - | 0.709 | - |
| Human[1] | 0.803 | 0.770 | 0.905 | 0.868 |
| Human (ours) | 0.650 | - | 0.807 | - |

| *NewsQA* | Exact Match | | F1 | |
| --- | --- | --- | --- | --- |
| Model | Dev | Test | Dev | Test |
| Random | 0.00 | 0.00 | 0.30 | 0.30 |
| mLSTM | 0.344 | 0.349 | 0.496 | 0.500 |
| BARB | 0.361 | 0.341 | 0.496 | 0.482 |
| Human | 0.465 | - | 0.694 | - |

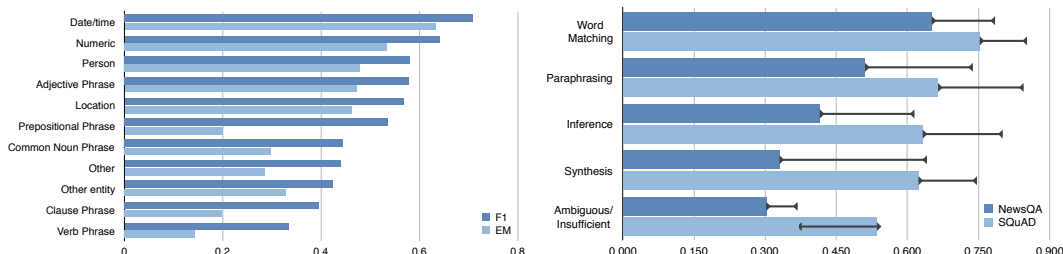

Figure 1: *Left*: BARB performance (F1 and EM) stratified by answer type on the full development set of *NewsQA*. *Right*: BARB performance (F1) stratified by reasoning type on the human-assessed subset on both *NewsQA* and *SQuAD*. Error bars indicate performance differences between BARB and human annotators.

the model is better at pointing to named entities compared to other types of answers. The reasoning-type stratification, on the other hand, shows that questions requiring *inference* and *synthesis* are, not surprisingly, more difficult for the model. Consistent with observations in Table 3, stratified performance on *NewsQA* is significantly lower than on *SQuAD*. The difference is smallest on word matching and largest on synthesis. We postulate that the longer stories in *NewsQA* make synthesizing information from separate sentences more difficult, since the relevant sentences may be farther apart. This requires the model to track longer-term dependencies.

## 6.3 SENTENCE-LEVEL SCORING

We propose a simple sentence-level subtask as an additional quantitative demonstration of the relative difficulty of *NewsQA*. Given a document and a question, the goal is to find the sentence containing the answer span. We hypothesize that simple techniques like word-matching are inadequate to this task owing to the more involved reasoning required by *NewsQA*.

We employ a technique that resembles inverse document frequency (*idf*), which we call inverse sentence frequency (*isf*). Given a sentence $\mathcal{S}_i$ from an article and its corresponding question $\mathcal{Q}$, the *isf* score is given by the sum of the *idf* scores of the words common to $\mathcal{S}_i$ and $\mathcal{Q}$ (each sentence is treated as a document for the *idf* computation). The sentence with the highest *isf* is taken as the answer sentence $\mathcal{S}_*$, that is,

$$\mathcal{S}_* = \arg\max_i \sum_{w \in \mathcal{S}_i \cap \mathcal{Q}} isf(w).$$

The *isf* method achieves an impressive 79.4% sentence-level accuracy on *SQuAD*'s development set but only 35.4% accuracy on *NewsQA*'s development set, highlighting the comparative difficulty of the latter. To eliminate the difference in article length as a possible cause of the performance difference, we also artificially increased the article lengths in *SQuAD* by concatenating adjacent *SQuAD* articles from the same Wikipedia document. Accuracy decreases as expected with the increased *SQuAD* article length, yet remains significantly higher than that on *NewsQA* with comparable or even larger article length (Table 4).

Table 4: Sentence-level accuracy on artificially-lengthened *SQuAD* documents.

|  | *SQuAD* | | | | | *NewsQA* |
|---|---|---|---|---|---|---|
| # documents | 1 | 3 | 5 | 7 | 9 | 1 |
| Avg # sentences | 4.9 | 14.3 | 23.2 | 31.8 | 40.3 | 30.7 |
| *isf* | 79.6 | 74.9 | 73.0 | 72.3 | 71.0 | 35.4 |

## 7 CONCLUSION

We have introduced a challenging new comprehension dataset: *NewsQA*. We collected the 100,000+ examples of *NewsQA* using teams of crowdworkers, who variously read CNN articles or highlights, posed questions about them, and determined answers. Our methodology yields diverse answer types and a significant proportion of questions that require some reasoning ability to solve. This makes the corpus challenging, as confirmed by the large performance gap between humans and deep neural models (0.198 in F1). By its size and complexity, *NewsQA* makes a significant extension to the existing body of comprehension datasets. We hope that our corpus will spur further advances in machine comprehension and guide the development of literate artificial intelligence.

## ACKNOWLEDGMENTS

The authors would like to thank Çağlar Gülçehre, Sandeep Subramanian and Saizheng Zhang for helpful discussions, and Pranav Subramani for the graphs.

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

APPENDICES

## A    IMPLEMENTATION DETAILS

Both mLSTM and BARB are implemented with the Keras framework (Chollet, 2015) using the Theano (Bergstra et al., 2010) backend. Word embeddings are initialized using GloVe vectors (Pennington et al., 2014) pre-trained on the 840-billion *Common Crawl* corpus. The word embeddings are not updated during training. Embeddings for out-of-vocabulary words are initialized with zero.

For both models, the training objective is to maximize the log likelihood of the boundary pointers. Optimization is performed using stochastic gradient descent (with a batch-size of 32) with the ADAM optimizer (Kingma & Ba, 2015). The initial learning rate is 0.003 for mLSTM and 0.0005 for BARB. The learning rate is decayed by a factor of 0.7 if validation loss does not decrease at the end of each epoch. Gradient clipping (Pascanu et al., 2013) is applied with a threshold of 5.

Parameter tuning is performed on both models using `hyperopt`[5]. For each model, configurations for the best observed performance are as follows:

**mLSTM**

Both the pre-processing layer and the answer-pointing layer use bi-directional RNN with a hidden size of 192. These settings are consistent with those used by Wang & Jiang (2016b).

Model parameters are initialized with either the normal distribution ($\mathcal{N}(0, 0.05)$) or the orthogonal initialization ($\mathcal{O}$, Saxe et al. 2013) in Keras. All weight matrices in the LSTMs are initialized with $\mathcal{O}$. In the Match-LSTM layer, $W^q$, $W^p$, and $W^r$ are initialized with $\mathcal{O}$, $b^p$ and $w$ are initialized with $\mathcal{N}$, and $b$ is initialized as 1.

In the answer-pointing layer, $V$ and $W^a$ are initialized with $\mathcal{O}$, $b^a$ and $v$ are initialized with $\mathcal{N}$, and $c$ is initialized as 1.

**BARB**

For BARB, the following hyperparameters are used on both *SQuAD* and *NewsQA*: $d = 300$, $D_1 = 128$, $C = 64$, $D_2 = 256$, $w = 3$, and $n_f = 128$. Weight matrices in the GRU, the bilinear models, as well as the boundary decoder ($\mathbf{v_s}$ and $\mathbf{v_e}$) are initialized with $\mathcal{O}$. The filter weights in the boundary decoder are initialized with *glorot_uniform* (Glorot & Bengio 2010, default in Keras). The bilinear biases are initialized with $\mathcal{N}$, and the boundary decoder biases are initialized with 0.

## B    DATA COLLECTION USER INTERFACE

Here we present the user interfaces used in question sourcing, answer sourcing, and question/answer validation.

---

[5]`https://github.com/hyperopt/hyperopt`

**Highlights**
- Three women to jointly receive the 2011 Nobel Peace Prize
- Prize recognizes non-violent struggle of safety of women and women's rights
- Prize winners to be honored with a concert on Sunday hosted by Helen Mirren

**Q1:**

Who were the prize winners?

**Q2:**

What country were the prize winners from?

**Q3:**

Write a question that relates to a highlight.

**Question**

What is the age of Patrick McGoohan?

☐ Click here if the **question does not make sense** or is **not a question**.

**Story**

(CNN) -- Emmy-winning **Patrick McGoohan,** the actor who created one of British television's most surreal thrillers, has died aged 80, according to British media reports.

Fans holding placards of **Patrick McGoohan** recreate a scene from 'The Prisoner' to celebrate the 40th anniversary of the show in 2007.

The Press Association, quoting his son-in-law Cleve Landsberg, reported he died in Los Angeles after a short illness.

**McGoohan,** star of the 1960s show 'The Danger Man,' is best remembered for writing and starring in 'The Prisoner' about a former spy locked away in an isolated village who tries to escape each episode.

**Question**

When was the lockdown initiated?

**Select the best answer:**
- ○ Tucson, Arizona,
- ◉ 10:30 a.m. --
- ○ 11 a.m.,
- ○ * All answers are very bad.
- ○ * The question doesn't make sense.

**Story (for your convenience)**

(CNN) -- U.S. Air Force officials called off their response late Friday afternoon at a Tucson, Arizona, base after reports that an armed man had entered an office building, the U.S. military branch said in a statement. Earlier in the day, a U.S. military official told CNN that a gunman was believed to be holed up in a building at the Davis-Monthan Air Force Base. This precipitated the Air Force to call for a lock-down -- which began at **10:30 a.m.** -- "following the unconfirmed sighting of" such a man. No shots were ever fired and law enforcement teams are on site, said the official, who had direct knowledge of the situation from conversations with base officials but did not want to be identified. In fact, at 6 p.m., Col. John Cherrey -- who commands the Air Force's 355th Fighter Wing -- told reporters that no gunman or weapon was ever found. He added that the building, where the gunman was once thought to

Figure 2: Examples of user interfaces for question sourcing, answer sourcing, and validation.

## Write Questions From A Summary

Instructions ▲

### Overview

Write questions about the highlights of a story.

### Steps

1. **Read** the **highlights**
2. **Write questions** about the highlights

### Example

**Highlights**

- Sarah Palin from Alaska meets with McCain
- Fareed Zakaria says John McCain did not put country first with his choice
- Zakaria: This is "hell of a time" for Palin to start thinking about national, global issues

**Questions**

The questions can refer directly to the highlights, for example:

- Where is Palin from?
- What did Fareed say about John McCain's choice?
- Who is thinking about global issues?

Questions **must** always be related to the highlights but their answers don't have to be in the highlights. You can assume that the highlights summarize a document which can answer other questions for example:

- What was the meeting about?
- What was McCain's choice?
- What issues is Palin thinking about?

### Other Rules

- Do **not** re-use the same or very similar questions.
- Questions should be written to have **short answers**.
  - Do **not** write "**how**" nor "**why**" type questions since their answers are not short. "How far/long/many/much" are okay.

Figure 3: Question sourcing instructions for the crowdworkers.

