# Peer review of "NEWSQA: A MACHINE COMPREHENSION DATASET"

_ICLR 2017 — rejected_

[Official Review · AnonReviewer1 · rating 6 · confidence 3 · 16 Dec 2016]
**Need better human evaluation and comparison with SQuAD**

Summary: The paper proposes a novel machine comprehension dataset called NEWSQA. The dataset consists of over 100,000 question answer pairs based on over 10,000 news articles from CNN. The paper analyzes the different types of answers and the different types of reasoning required to answer questions in the dataset. The paper evaluates human performance and the performance of two baselines on the dataset and compares them with the performance on SQuAD dataset. 

Strengths:

1. The paper presents a large scale dataset for machine comprehension. 

2. The question collection method seems reasonable to collect exploratory questions. Having an answer validation step is desirable.

3. The paper proposes a novel (computationally more efficient) implementation of the match-LSTM model.

Weaknesses:

1. The human evaluation presented in the paper is not satisfactory because the human performance is reported on a very small subset (200 questions). So, it seems unlikely that these 200 questions will provide a reliable measure of the human performance on the entire dataset (which consists of thousands of questions).

2. NEWSQA dataset is very similar to SQuAD dataset in terms of the size of the dataset, the type of dataset -- natural language questions posed by crowdworkers, answers comprising of spans of text from related paragraphs. The paper presents two empirical ways to show that NEWSQA is more challenging than SQuAD -- 1) the gap between human and machine performance in NEWSQA is larger than that in SQuAD. However, since the human performance numbers are reported on very small subset, these trends might not carry over when human performance is computed on all of the dataset.
2) the sentence-level accuracy on SQuAD is higher than that in NEWSQA. However, as the paper mentions, the differences in accuracies could likely be due to different lengths of documents in the two datasets. So, even this measure does not truly reflect that SQuAD is less challenging than NEWSQA.
So, it is not clear if NEWSQA is truly more challenging than SQuAD.

3. Authors mention that BARB is computationally more efficient and faster compared to match-LSTM. However, the paper does not report how much faster BARB is compared to match-LSTM.

4. On page 7, under "Boundary pointing" paragraph, the paper should clarify what "s" in "n_s" refers to.

Review summary: While the dataset collection method seems interesting and promising, I would be more convinced after I see the following --
1. Human performance on all (or significant percentage of the dataset).
2. An empirical study that fairly shows that NEWSQA is more challenging (or better in some other way) than SQuAD.

[Official Review · AnonReviewer2 · rating 6 · confidence 4 · 17 Dec 2016]
**potentially great dataset with some flaws**

It would seem that the shelf life of a dataset has decreased rapidly in recent literature. SQuAD dataset has been heavily pursued as soon as it hit online couple months ago, the best performance on their leaderboard now reaching to 82%. This is rather surprising when taking into account the fact that the formal conference presentation of the dataset took place only a month ago at EMNLP’16, and that the reported machine performance (at the time of paper submission) was only at 51%. One reasonable speculation is that the dataset may have not been hard enough.

NewsQA, the paper in submission, aims to address this concern by presenting a dataset of a comparable scale created through different QA collection strategies. Most notably, the authors solicit questions without requiring answers from the same turkers, in order to promote more diverse and hard-to-answer questions. Another notable difference is that the questions are gathered without showing the content of the news articles, and the dataset makes use of a bigger subset of CNN/Daily corpus (12K / 90K), as opposed to a much smaller subset (500 / 90K) used by SQuAD.

In sum, I think NewsQA dataset presents an effort to construct a harder, large-scale reading comprehension challenge, a recently hot research topic for which we don’t yet have satisfying datasets. While not without its own weaknesses, I think this dataset presents potential values compared to what are available out there today.

That said, the paper does read like it was prepared in a hurry, as there are numerous small things that the authors could have done better. As a result, I do wonder about the quality of the dataset. For one, human performance of SQuAD measured by the authors (70.5 - 82%) is lower than that reported by SQuAD (80.3 - 90.5%). I think this sort of difference can easily happen depending on the level of carefulness the annotators can maintain. After all, not all humans have the same level of carefulness or even the same level of reading comprehension. I think it’d be the best if the authors can try to explain the reason behind these differences, and if possible, perform a more careful measurement of human performance. If anything, I don’t think it looks favorable for NewsQA if the human performance is only at the level of 74.9%, as it looks as if the difficulty of the dataset comes mainly from the potential noise from the QA collection process, which implies that the low model performance could result from not necessarily because of the difficulty of the comprehension and reasoning, but because of incorrect answers given by human annotators.

I’m also not sure whether the design choice of not presenting the news article when soliciting the questions was a good one. I can imagine that people might end up asking similar generic questions when not enough context has been presented. Perhaps taking a hybrid, what I would like to suggest is to present news articles where some sentences or phrases are randomly redacted, so that the question generators can have a bit more context while not having the full material in front of them.

Yet another way of encouraging the turkers from asking too trivial questions is to engage an automatic QA system on the fly — turkers must construct a QA pair for which an existing state-of-the-art system cannot answer correctly.

[Official Review · AnonReviewer3 · rating 6 · confidence 4 · 17 Dec 2016]

Paper Summary: 
This paper presents a new comprehension dataset called NewsQA dataset, containing 100,000 question-answer pairs from over 10,000 news articles from CNN. The dataset is collected through a four-stage process -- article filtering, question collection, answer collection and answer validation. Examples from the dataset are divided into different types based on answer types and reasoning required to answer questions. Human and machine performances on NewsQA are reported and compared with SQuAD.

Paper Strengths: 
-- I agree that models can benefit from diverse set of datasets. This dataset is collected from news articles, hence might pose different sets of problems from current popular datasets such as SQuAD.
-- The proposed dataset is sufficiently large for data hungry deep learning models to train. 
-- The inclusion of questions with null answers is a nice property to have.
-- A good amount of thought has gone into formulating the four-stage data collection process.
-- The proposed BARB model is performing as good as a published state-of-the-art model, while being much faster.    

Paper Weaknesses: 
-- Human evaluation is weak. Two near-native English speakers' performance on 100 examples each can hardly be a representative of the complete dataset. Also, what is the model performance on these 200 examples?
-- Not that it is necessary for this paper, but to clearly demonstrate that this dataset is harder than SQuAD, the authors should either calculate the human performance the same way as SQuAD or calculate human performances on both NewsQA and SQuAD in some other consistent manner on large enough subsets which are good representatives of the complete datasets. Dataset from other communities such as VQA dataset (Antol et al., ICCV 2015) also use the same method as SQuAD to compute human performance. 
-- Section 3.5 says that 86% of questions have answers agreed upon by atleast 2 workers. Why is this number inconsistent with the 4.5% of questions which have answers without agreement after validation (last line in Section 4.1)?
-- Is the same article shown to multiple Questioners? If yes, is it ensured that the Questioners asking questions about the same article are not asking the same/similar questions?
-- Authors mention that they keep the same hyperparameters as SQuAD. What are the accuracies if the hyperparameters are tuned using a validation set from NewsQA?
-- 500 examples which are labeled for reasoning types do not seem enough to represent the complete dataset. Also, what is the model performance on these 500 examples?
-- Which model's performance has been shown in Figure 1?
-- Are the two "students" graduate/undergraduate students or researchers?
-- Test set seems to be very small.
-- Suggestion: Answer validation step is nice, but maybe the dataset can be released in 2 versions -- one with all the answers collected in 3rd stage (without the validation step), and one in the current format with the validation step. 

Preliminary Evaluation: 
The proposed dataset is a large scale machine comprehension dataset collected from news articles, which in my suggestion, is diverse enough from existing datasets that state-of-the-art models can definitely benefit from it. With a better human evaluation, I think this paper will make a good poster.

[Author Response · Tong Wang · 04 Jan 2017]
**Quick update**

Dear readers and reviewers,                                                                                                                                                                                                                   
  Happy holidays! Before the final revision, we'd like to give you a quick update on our latest efforts and progress. We have currently completed 95% of the planned expansion on human evaluations (on both NewsQA and SQuAD), and we're in the process of hyperparameter tuning for both MatchLSTM and our BARB model on NewsQA. Our plan is to update relevant statistics and results no later than the end of next week.                                                                         
  Thank you!

[Author Response · Tong Wang · 16 Jan 2017]
**Expanded human evaluation, more sentence-level experiments, etc.**

Thank you everyone for your patience. Here's a list of updates we've made per the comments and suggestions of our reviewers.
Major updates:
* Increased our human evaluation of NewsQA from 100 to 1000 samples with reasoning type annotation.
* Performed the same human evaluation on 1000 SQuAD with reasoning type annotation.
* Updated stratification results (by reasoning type) for both NewsQA and SQuAD on the expanded subsets evaluated by human annotators.
* Added more sentence-level scoring experiments for a more conclusive demonstration of NewsQA's increased level of difficulty over SQuAD (eliminating article length as a factor in performance difference).
* Performed hyper-parameter tuning for both BARB and MatchLSTM model on the NewsQA dataset.
Minor updates:
* Added machine performance on the human-evaluated subsets for both NewsQA and SQuAD.
* Updated various statistics to be consistent with NewsQA's public release.
* Various minor updates, corrections, and additional statistics per requested in the reviews.

Thank you.

[Final Decision · Program Chairs · 06 Feb 2017]
**ICLR committee final decision**

The program committee appreciates the authors' response to concerns raised in the reviews. All reviewers agree that the paper is not convincingly above the acceptance threshold. The paper will be stronger, and the benefit of this dataset over SQuAD will likely be more clear once authors incorporate reviewers' comments and finish evaluation of inter-human agreement.